# Economic Evaluation of Glucosamine in Knee Osteoarthritis Treatments in Vietnam

**DOI:** 10.3390/healthcare11182502

**Published:** 2023-09-08

**Authors:** Nam Xuan Vo, Uyen Thi Thuc Che, Thanh Thi Thanh Ngo, Tien Thuy Bui

**Affiliations:** 1Faculty of Pharmacy, Ton Duc Thang University, Ho Chi Minh City 700000, Vietnam; thucuyenpct@gmail.com (U.T.T.C.); thahthah0906@gmail.com (T.T.T.N.); 2Faculty of Pharmacy, Le Van Thinh Hospital, Ho Chi Minh City 700000, Vietnam; thuytienbui2404@gmail.com

**Keywords:** economic evaluation, cost-utility analysis, glucosamine, knee osteoarthritis, Vietnam

## Abstract

Osteoarthritis (OA) is the degeneration of cartilage in joints that results in bones rubbing against each other; it causes uncomfortable symptoms such as pain, swelling, and stiffness and can lead to disability. It usually occurs in the elderly and causes a large medical burden. The aim of this study is to evaluate the cost-effectiveness between the standard treatment for osteoarthritis and standard treatment with added crystalline glucosamine sulfate at various stages. Markov analysis modeling was applied to evaluate the effect of both adding glucosamine compared to standard treatment from a societal perspective during whole patients’ lifetimes. Data input was collected from reviews in previous studies. The outcome was measured in quality-adjusted life years (QALYs), and the Incremental Cost-Effectiveness Ratio (ICER) from a societal perspective was applied with 3% and discounted for all costs and outcomes. One-way analysis via the Tornado diagram was performed to investigate the change in factors in the model. In general, adding glucosamine into the standard treatment proved to be more cost-effective compared to the standard treatment. Particularly, the early-stage addition of glucosamine in the treatment was cost-effective compared to the post-stage addition of glucosamine. The addition of supplementing crystalline glucosamine sulfate to the whole regimen at any stage was cost-effective at the willingness-to-pay (WTP) threshold.

## 1. Introduction

Osteoarthritis (OA) has emerged as a common chronic musculoskeletal disorder, becoming a common medical condition affecting the joints. In particular, osteoarthritis affects all races, sexes, and ages but is most commonly seen in obese and elderly people [1]. Osteoarthritis has multiple risk factors, which can be categorized into two main types: modifiable risk factors and nonmodifiable risk factors. Modifiable risk factors include previous joint injury, obesity, metabolic syndrome, and daily lifestyle factors (such as occupational stress or overexertion). Nonmodifiable risk factors encompass genetics, increasing with age and the female gender. People with risk factors should undergo regular health check-ups to enable timely interventions during the early stages and ensure the best treatment approach, thus avoiding potential long-term consequences [1,2,3,4,5].

In the human body, the meniscus is known to facilitate joint stability, maintain normal knee function, lubricate joints, and distribute body loads [6,7,8]. Osteoarthritis develops when the protective cartilage covering the ends of bones gradually wears down, leading to the bones rubbing against each other. This friction causes common symptoms such as joint swelling, pain, and stiffness [4,5]. In general, the partial or complete loss of function of the meniscus causes knee osteoarthritis [9,10]. Various imaging techniques can aid in diagnosing OA and providing treatment recommendations. Plain radiographs (X-rays) are commonly used and can show characteristic features of OA, including joint space narrowing, osteophytosis (bone spurs), subchondral sclerosis (increased bone density beneath the cartilage), and cyst formation [4,5]. Diagnostic ultrasounds and MRIs can also be employed to visualize soft tissues and assess the severity of joint damage [4,5,8]. Osteoarthritis can affect various joints throughout the body, including the wrist and finger joints, ankle and toe joints, hip joints, the cervical spine, and the lumbar spine. However, the most commonly affected joint is the knee joint. Knee arthritis includes self-reported knee arthritis, the radiographic definition of knee arthritis, and symptomatic knee arthritis (self-reported joint pain, stiffness, pain, and radiographic evidence) [11].

Worldwide, osteoarthritis is considered the fourth leading cause of disability and functional impairment [12,13]. In developed and developing countries, osteoarthritis can cause a significant deterioration in the quality of life for people over 65 years of age due to joint pain and disability [3,14,15,16,17]. In an earlier Australian study, in Caucasians, the prevalence of symptomatic osteoarthritis was about 10% in men and 20% in women aged 45 years and older [18], but for radiographic OA, the incidence could be increased in the range from 27% to 80% in research conducted in the USA [19]. In addition, the population prevalence of knee osteoarthritis at 40 years old and above accounts for 34%, with a higher incidence rate among females (62%) compared to males (35%) in Ho Chi Minh City, Vietnam [20]. The treatment of osteoarthritis is increasingly becoming a burden on society, not only in terms of direct healthcare costs but also regarding indirect costs resulting from the rising prevalence of the disease coupled with population aging [21]. This is a significant public health issue that requires attention as this disease leads to functional impairment and becomes an economic burden on society [16].

Currently, interventions for patients with osteoarthritis include lifestyle modifications, medication-based approaches, and surgical interventions. lifestyle modifications, such as maintaining a healthy weight, engaging in low-impact exercises, protecting joints from injuries, avoiding repetitive joint stress, following a balanced diet, and managing overall health conditions, can significantly reduce the risk of developing osteoarthritis and promote optimal joint health [22]. Medication-based approaches include pain relievers, symptomatic slow-acting drugs for osteoarthritis (SYSADOAs), non-steroidal anti-inflammatory drugs (NSAIDs), and intra-articular corticosteroid injections. Surgical interventions involve partial or total joint replacement [23]. In reality, non-steroidal anti-inflammatory drugs (NSAIDs) are the most commonly used medication due to their effectiveness in reducing pain and inflammation. However, they do carry a significant risk of adverse effects, such as cardiovascular issues and gastrointestinal systems [24].

Glucosamine is a SYSADOA medication prescribed for patients with mild to moderate osteoarthritis. It plays a role in the synthesis and metabolism of joint cartilage, promoting the production of essential components and increasing the production of synovial fluid, which enhances lubrication. As a result, it effectively reduces the symptoms of disease, such as swelling, pain, and joint stiffness, while also slowing down its progression [2]. Glucosamine offers a cost-effective treatment option with favorable outcomes [25]. The glucosamine sulfate crystalline form, in particular, has been proven to be more cost-effective in the treatment of joint inflammation compared to other formulations. By incorporating glucosamine into treatment plans, healthcare providers can provide patients with an effective therapy that not only alleviates symptoms but also helps prevent the further deterioration of the condition. This leads to improved overall treatment outcomes at a lower cost. The use of glucosamine as a therapeutic option highlights its potential in managing osteoarthritis and its cost-effectiveness compared to alternative treatments [26]. Its ability to target the underlying processes of joint degeneration and provide symptomatic relief makes it a valuable addition to the treatment regimen for patients with mild to moderate osteoarthritis [2,27,28]. Some studies have shown that the use of glucosamine in the treatment of osteoarthritis is effective. Still, these effects have only been proven through studies over a short period. The long-term effectiveness of glucosamine use has not been established [27,28,29]. Because the clinical effectiveness of glucosamine, as demonstrated in published research studies, remains inconclusive, this has led to significant debates and difficulties in the decision making of reimbursement agencies [29,30,31,32,33].

The issue at hand is not only finding a highly effective and cost-efficient treatment method but, more importantly, identifying the most efficient treatment sequence for osteoarthritis while minimizing costs and reducing the potential adverse effects of medications.

Research has also aimed to identify the optimal integration of crystalline glucosamine sulfates into standardized treatment following the most favorable sequence. Through this research, we aim to provide healthcare professionals and policymakers with a comprehensive framework that can evaluate and compare treatment regimens, allowing them to make informed decisions based on cost-effective analysis.

## 2. Materials and Methods

A comprehensive economic evaluation was conducted, considering both the clinical outcomes and associated costs. This study employed a Markov model with standard treatment 1 (including pain relievers, one type of NSAID, intra-articular corticosteroid injections, and surgery) and standard treatment 2 (including pain relievers, two types of NSAIDs, intra-articular corticosteroid injections, and surgery) as the benchmarks to compare cost-effectiveness with the protocol supplemented in crystalline glucosamine sulfate at different stages. The study participants consisted of a group of mild to moderate osteoarthritis patients above the age of 40 without any severe accompanying conditions (cardiovascular, diabetes, etc.). The study period spanned the lifetime horizon of individuals, lasting for a 6-month cycle length. Data were derived from a comprehensive review of the literature based on publicly available databases and data from the Vietnam Drug Administration (DAV). This evaluation of effectiveness was based on quality-adjusted life years (QALYs), which adjusted the additional years of life gained for the quality of life, and cost-effectiveness was assessed using the Incremental Cost-Effectiveness Ratio (ICER) from a societal perspective. All costs and outcomes in this study were discounted at 3%.

ICER (Incremental Cost-Effectiveness Ratio) is an essential tool in health economics that assists policymakers in evaluating the efficiency and rationality of different treatment sequences based on their cost and benefits. In the research model, ICER is calculated by comparing the costs and quality-adjusted life years (QALYs) of each treatment sequence.
ICER = ΔCost/ΔQALY (incremental costs/incremental QALY gained)

### 2.1. Model Structure

This study utilized the Markov model to evaluate and compare the cost-effectiveness of treatment regimens against standard treatments. The methodology involved constructing models that represented various treatment pathways and their corresponding probabilities, costs, and outcomes. By considering factors such as treatment efficacy, adverse effects, and costs, this model can provide valuable insights into the cost-effectiveness of different treatment regimens. This model was reviewed and consulted with local experts.

The standard treatment regimen in this study adhered to the following treatment sequence:

Standard treatment 1 (PD): pain relievers (Acetaminophen), non-steroidal anti-inflammatory drugs (NSAIDs): Diclofenac combined with a proton pump inhibitor (PPI), intra-articular corticosteroid injection (Triamcinolone) and finally, total arthroplasty surgery.

Standard treatment 2 (PDE): Standard treatment 1 + Etoricoxib. The specific treatment sequence was as follows: Acetaminophen-Diclofenac + PPI-Etoricoxib–Triamcinolone-total arthroplasty surgery.

Taking the milestone as standard treatment 1 and standard treatment 2, in each standard regimen, crystalline glucosamine sulfate was supplemented before and after NSAIDs, forming different treatment sequences for the cost-effectiveness comparison. In treatment 1, glucosamine was used before initiating the use of NSAIDs (non-steroidal anti-inflammatory drugs). Glucosamine is commonly used to alleviate pain and slow down the progression of joint degeneration, while NSAIDs are typically used to reduce inflammation and pain. By using glucosamine before NSAIDs, the objective was to enhance the effectiveness of slowing down disease progression and reduce the necessity of NSAIDs during the treatment process. In treatment 2, glucosamine was used after the administration of NSAIDs. The objective was to evaluate the differences in the extent of disease progression retardation and the cost-effectiveness of treatment when glucosamine was utilized as a supplementary approach following the use of NSAIDs. Utilizing glucosamine after NSAIDs could provide additional benefits in slowing down the progression of joint degeneration after inflammation and pain reduction with NSAIDs. The disparity between these two treatment sequences compared to the standard treatment protocol could lie in the effectiveness of slowing down joint degeneration, the level of pain and inflammation reduction, as well as the cost and convenience of utilizing these methods. However, to obtain an accurate assessment of the differences between these two treatment sequences, an evaluation based on specific research data is required. The detail regimens were showed in Figure 1.

Regimen 1 (PGD): the addition of crystalline glucosamine sulfate before Diclofenac + PPI in standard treatment 1; Regimen 2 (PDG): the addition of crystalline glucosamine sulfate after Diclofenac + PPI in standard treatment 1.

Regimen 3 (PGDE): the addition of crystalline glucosamine sulfate before Diclofenac + PPI + Etoricoxib in standard treatment 2; Regimen 4 (PDEG): the addition of crystalline glucosamine sulfate after Diclofenac + PPI+ Etoricoxib in standard treatment 2.

In each cycle of the model, patients could achieve a stable state with the same level of treatment or transition to the next treatment state when the medication was unresponsive while pain remained unimproved, necessitating intervention with escalated treatment measures. Patients in the study commenced treatment with Acetaminophen and ended in a deceased state. The transition rate, which represents the conversion rate between stages within each cycle of the model, was utilized to calculate the number of quality-adjusted life years (QALYs) in each phase. The main assumption of this study was that patients only received the prescribed treatment modality in each specific state.

### 2.2. Model Assumptions

The main assumption in our study is that patients were only treated as prescribed by doctors at certain stages, as in the proposed model. The risks of possible complications and side effects were assumed to be unchanged over time. The patients who received treatment underwent recovery or were moved to the next health stage. Once the patient moved to the next stage of illness, it was not possible to return to the previous state of illness.

Regarding the literature review of economic evaluation studies for osteoarthritis of the knee, all studies, including ours, hypothesized that self-treatment or an adverse effects disease subtype would only be considered as a disease stage in our model. This is due to the nature of osteoarthritis, where it is difficult to define specific stages; therefore, adverse events were not included.

In Vietnam, the majority of people who begin to have symptoms and are recorded as having osteoarthritis are at the age of 40 or older: the mean age to catch osteoarthritis is 55 years old [20]. Therefore, the target population selected for this study was 40 years old and older. In addition, knee osteoarthritis is a chronic disease, so in this study, QALYs based on the status of patients were estimated as the health outcome of this model. The life years gained were estimated to be equal to the number of years of life that a person from 40 years old had compared to the average life expectancy of Vietnamese people in 2021 [34].

### 2.3. Model Input

Data were collected through a review of the literature based on studies available in Vietnam. The missing data were searched and summarized through world studies, i.e., meta-analysis, systematic review, randomized controlled trial (RCT), and real-world evidence studies. The data were selected from data in studies from countries with the same socio-economic conditions as Vietnam or with countries similar to Vietnam, such as Thailand. The 3% discount rate was applied to both costs and outcomes. The data are summarized in Table 1.

### 2.4. Cost Variables

The cost data included for calculation comprised:

Direct medical costs: These encompass expenses related to direct healthcare services such as medication costs, laboratory tests, consultations, and hospital bed utilization during the treatment of osteoarthritis. Medication costs within the model were computed based on the maximum dosage per patient based on the Ministry of Health’s diagnostic and treatment guidelines.

Direct non-medical costs: These encompass expenses incurred by patients that are not directly related to healthcare services but are associated with the treatment and management of osteoarthritis. This includes costs for meals, transportation, and travel to healthcare facilities [43]. The Consumer Price Index (CPI) of Vietnam was used to convert past monetary values to the present [49].

Indirect costs: These include costs incurred due to work absenteeism and reduced productivity caused by pain. Pain and functional limitations can significantly impact the patient’s ability to work effectively, leading to a loss of income and decreased work productivity. In this study, costs were estimated from knee joint recovery post-surgery multiplied by Vietnam’s Gross Domestic Product (GDP) per capita per month. The knee joint recovery post-surgery (1 month) was estimated from Hien TT. et al., 2022 [44].

According to the World Bank, Vietnam’s Gross Domestic Product per capita in 2021 was USD 3756 [50]. Based on a currency exchange rate of USD 1, equivalent to VND 23,486 (May 2023), Vietnam’s GDP per capita in 2021 would be approximately VND 88,213,416 [50]. At present, the willingness to pay the threshold in Vietnam is determined based on per capita GDP. When the ICER (Incremental Cost-Effectiveness Ratio) is lower than the per capita GDP, the expenditure is considered highly cost-effective. When the ICER is greater than 1 GDP but less than 3 times the per capita GDP, the expenditure is considered cost-effective. If the ICER exceeds 3 times the per capita GDP, the expenditure is deemed not cost-effective.

### 2.5. Efficacy of Pain Relief

All efficacy data were assumed through data collected from RCTs worldwide because of limited data available in Vietnam. From selected studies [35,36,37,38,39], data on the clinical effectiveness of a group of patients with osteoarthritis exhibiting characteristics relevant to the Vietnamese population were gathered. The collected results regarding pain reduction efficacy were transformed into transition rates for utilization within the model’s calculations. Specifically, the clinical effectiveness of Acetaminophen, Diclofenac + PPI, Etoricoxib, crystalline glucosamine sulfate, Triamcinolone injection, and total arthroplasty were as follows for previous studies.

### 2.6. Adverse Events

While commonly used for pain and joint-related issues, Acetaminophen, Diclofenac, Etoricoxib, and Triamcinolone injections have potential adverse effects that should be considered. Acetaminophen, although generally considered safe when used as directed, can cause gastrointestinal disturbances such as stomach upsets. In high doses or with prolonged use, it can lead to liver toxicity, which poses a significant concern. Although rare, allergic reactions to Acetaminophen have been reported. Diclofenac and Etoricoxib, a non-steroidal anti-inflammatory drug (NSAID), carry their own set of adverse effects. Gastrointestinal ulcers or bleeding can occur, particularly with long-term use or high doses. Furthermore, Diclofenac is associated with cardiovascular risks, including an increased likelihood of high blood pressure, heart attack, or stroke. Renal toxicity and allergic reactions, manifesting as a skin rash, swelling, or difficulty breathing, are also possible adverse effects [39]. In the case of Triamcinolone injection, local injection site reactions, such as pain, swelling, or infection, are possible. There may be a temporary flare-up of joint pain or inflammation following the injection [37]. In this study, adverse events are not included.

### 2.7. Health Outcomes

The utility of each health state was assessed based on the effectiveness of the respective medications in alleviating pain. The utility index score of oral medications was obtained from Latimer et al. (2009) [45]. The utility of patients using glucosamine was retrieved from the study of Olivier Bruyère et al. (2019) [46]. For joint replacement surgery, a utility index score of 0.76 was derived from David Feeny’s study (2004) [48]. The utility index score of 0.64 for injectable medications was sourced from Losina’s study (2013) [47], which obtained data from the Osteoarthritis Research Society International. The pain subscale from the Western Ontario and McMaster Universities OA Index (WOMAC) served as the primary measure of clinical efficacy [37].

### 2.8. Sensitivity Analysis

One-way analysis with results presented as a Tornado diagram was applied to analyze the effect of parameters on the ICER values when parameters values were changed.

## 3. Results

### 3.1. Base Case

#### 3.1.1. The Base-Case of Standard Treatment 1

Standard treatment 1 had a cost of VND 314,758,471 and a quality-adjusted life year (QALY) of 56.9374.

When glucosamine was added to the standard treatment 1 before using Diclofenac plus PPI, an increase in both cost and effectiveness was observed. The additional crystalline glucosamine sulfate treatment before NSAIDs had a cost of VND 348,133,295 and a QALY of 100.7445. This indicated that the supplementation of glucosamine increased the treatment cost compared to standard treatment 1 but also improved treatment effectiveness based on QALY. The cost difference between the additional treatment and the standard treatment was VND 33,374,824, while the QALY difference was 43.8071. To assess the economic efficiency of adding glucosamine, the Incremental Cost-Effectiveness Ratio (ICER) was calculated and defined as the cost difference divided by the QALY difference. The ICER of VND/QUALY 761,858 suggested a relatively moderate cost to achieve an additional QALY when using additional glucosamine treatment before NSAIDs compared to standard treatment 1. Moreover, it is noteworthy that the obtained ICER was lower than the threshold of 1 GDP per capita in Vietnam. This implies that the supplementation of glucosamine before NSAIDs could be considered economically efficient.

When supplementing crystalline glucosamine sulfate in standard treatment 1 (after Diclofenac plus PPI), the cost was VND 339,027,398 with a corresponding quality-adjusted life year (QALY) of 92.7119. Compared to standard treatment 1, there was a cost difference of VND 24,268,927 and a QALY difference of 35.7745. This indicates that additional glucosamine treatment after NSAIDs had a slightly higher cost but also improved the treatment effectiveness in terms of QALYs to a reasonable extent. The Incremental Cost-Effectiveness Ratio (ICER) was calculated to be 678,386 VND/QALY.

In addition, when comparing the administration of glucosamine before NSAIDs with its administration after NSAIDs, it was observed that the former had a slightly higher cost but also higher QALYs. The calculated ICER for this case was VND/QALY 1,133,615, indicating the cost needed to achieve an additional QALY.

#### 3.1.2. The Base Case of Standard Treatment 2

For standard treatment 2, which includes the addition of Etoricoxib compared to standard treatment 1, it was observed that the cost increased when Etoricoxib was introduced (VND 367,478,671). However, at the same time, the QALYs doubled, increasing from 56.9374 to 129.8038. The Incremental Cost-Effectiveness Ratio (ICER) for this comparison was VND/QULY 723,519.

The outcomes of incorporating glucosamine into standard treatment 2 showed similarities to those observed in standard treatment 1. Incorporating crystalline glucosamine sulfate into standard treatment 2 before using Diclofenac plus PPI led to notable increases in both cost and effectiveness. The addition of crystalline glucosamine sulfate before NSAIDs resulted in a cost of VND 443,969,715 and a QALY of 230.6720. This indicates that while the supplementation of glucosamine increased the treatment cost compared to standard treatment 2, it also significantly improved treatment effectiveness in terms of QALY. The cost difference between additional treatment and standard treatment amounted to VND 76,491,044, with a QALY difference of 100.8682 (a substantial increase in QALY). The calculated ICER was VND/QALY 758,326, and ICER was found to be lower than the threshold of 1 GDP per capita in Vietnam.

In addition, when adding crystalline glucosamine sulfate into standard treatment 2 (after NSAIDs), the treatment cost amounted to VND 425,013,733, and QALY was 211.3288. Compared to standard treatment 2, the cost difference was VND 57,535,061, accompanied by a QALY difference of 81.5250. This suggests that the additional glucosamine treatment incurred a slightly higher cost while yielding improved QALY outcomes. The resulting ICER was calculated at 705,735 VND/QALY.

Moreover, when comparing the addition of glucosamine before NSAIDs into the treatment versus the addition of glucosamine after NSAIDs, the QALYs decreased from 230.6720 to 211.3288. The ICER was 979,981 VND/QALY.

All the results are summarized in Table 2 below.

### 3.2. Sensitivity Analyses

Based on the analysis of one-way sensitivity using tornado diagrams in all four scenarios of adding Glucosamine to standard treatment 1 and standard treatment 2 when glucosamine was added before NSAIDs (Figure 2) and before NSAIDs (Figure 3), the factor that had the most significant impact on ICER was cost and the utility of total knee arthroplasty. This is due to the high cost of knee replacement and the relief of pain after total knee arthroplasty. The next factor affected by ICER was cost and the utility of Acetaminophen use and other oral medications. Almost all these medications help the patient with pain relief during the whole period of the disease.

This shows that the cost and efficacy of glucosamine have a substantial impact on the cost-effectiveness analysis compared to the standard treatment sequence. Furthermore, the cost and utility of the final treatment methods in the treatment sequence (Triamcinolone injection) had a minimal effect on the analysis results because the proportion of patients transitioning to advanced treatment methods was low. Earlier treatment methods have a higher proportion of patients using them, leading to a higher level of impact on ICER. Later advanced treatment methods in the sequence had a lower impact on ICER outcomes. One-way sensitivity also showed the same trend when the cost and utility of glucosamine in patients who used glucosamine before NSAIDs were better than those who used glucosamine after NSAIDs in osteoarthritis treatment.

## 4. Discussion

This study was conducted to evaluate the cost-effectiveness of using glucosamine in the treatment of osteoarthritis in Vietnam. The calculated ICER in all cases shows that glucosamine is cost-effective for osteoarthritis treatment in Vietnam. The ICER range in all the cases ranged from VND 700,000 (USD 29.8) to VND 1,140,000 (USD 48.5) per QALY, which was much lower when compared to 1 GDP in Vietnam. This happens when the cost of glucosamine in Vietnam is relatively cheap. The most influential factor for ICER is the cost of total arthroplasty and the quality-of-life index for total arthroplasty patients.

In order to identify an optimal treatment sequence for joint degeneration, it is crucial to analyze both the effects of symptom reduction, disease progression prevention, and the minimization of adverse effects while maintaining cost-effectiveness. The model serves as a valuable analytical tool to assess the cost-effectiveness of various treatment methods over an extended period of time. Based on the cost-effectiveness analysis conducted within the context of Vietnam’s readiness-to-pay threshold, all treatment sequences that involved the supplementation of crystalline glucosamine sulfate into the standard treatment demonstrated favorable cost-effectiveness. This suggests that incorporating crystalline glucosamine sulfate into the standard treatment for osteoarthritis could provide economic benefits compared to the standard treatment sequence.

According to research, this treatment sequence could affect the treatment outcomes of patients. Initiating the use of crystalline glucosamine sulfate before NSAIDs could be considered a dominant strategy in the treatment of osteoarthritis. In doing so, glucosamine demonstrates its benefits in reducing pain symptoms and improving joint function from the early stages of the disease, thereby enhancing the patient’s quality of life. This could help prevent the rapid progression of the disease and reduce the risk of more complex treatment methods, such as joint replacement surgery, a major and costly surgical procedure. One strength of crystalline glucosamine sulfate compared to other treatment methods is its lower incidence of side effects. While other treatment methods may cause adverse effects such as gastric ulcers as well as digestive and cardiovascular disturbances, glucosamines have fewer side effects and are considered safer in the majority of cases [33,51,52]. Furthermore, the early use of glucosamine can also help reduce the need for NSAIDs. This is significantly important as the prolonged use of NSAIDs can lead to adverse effects, especially for patients at high risk of gastrointestinal and cardiovascular issues [53].

The results from this study are consistent with the results from several previous studies on the cost-effectiveness of glucosamine, such as Scholtissen et al. (2010) [53], who demonstrated that flucosamine is cost-effectiveness compared to paracetamol and a placebo. Bruyère et al. (2019, 2021) [46,54] concluded that crystalline glucosamine sulfate is cost-effective compared to a placebo and other forms of glucosamine at 3-month, 6-month, and 3-year time points. Additionally, Segal et al. (2004) [55], B.D. Zhang et al. (2012) [26] and Luksameesate et al. (2022) [56] demonstrated the cost-effectiveness of glucosamine in the treatment of osteoarthritis. This indicates the feasibility of crystalline glucosamine sulfate to improve cost-effectiveness compared to other treatment methods. Furthermore, Black et al. (2009) [57] and Chaiyakunapruk N et al. (2010) [58] stated that glucosamine is clinically effective but does not achieve the desired cost-effectiveness. This observed difference could be attributed to varying state management policies and differences in the structure of the research model.

Currently, glucosamine is a licensed product in Vietnam. It is covered by health insurance. According to the results of this study, the inclusion of glucosamine in the early stages of standard treatment can be seen as a good alternative to reduce treatment costs and improve osteoarthritis patients’ health. This study provides additional evidence to assist physicians and policymakers in making appropriate decisions to ensure equal support for patients when needed.

The study has a number of limitations that should be considered. First, not all treatment methods and medications used in the management of joint degeneration were included in this study. The focus was primarily on commonly employed methods in clinical practice, which were considered appropriate for incorporation into the research model. Second, this study relied on a comprehensive review of the literature approach, which involved gathering information from various data sources. This was necessary due to the limited availability of data specific to Vietnam. Therefore, data from different countries were included to compensate for the lack of local data. However, it is important to note that this approach might introduce potential variations in terms of social conditions, lifestyles, and policies between countries. Additionally, this study selectively chose literature sources from neighboring countries with similarities in social context, lifestyles, and policies, which were deemed relevant for reference purposes. This approach aimed to provide valuable insights; however, it is crucial to acknowledge that the applicability of these findings to the specific context of Vietnam may be influenced by these variations. Moreover, the ICER did not distinguish between males and females. These limitations highlight the need for further research that encompasses a broader range of treatment methods, utilizing local data and considering the specific socio-cultural and policy contexts of Vietnam. Such studies could contribute to a more comprehensive understanding of the effectiveness and cost-effectiveness of treatments for joint degeneration in the Vietnamese population.

## 5. Conclusions

In conclusion, this study provides evidence that the addition of crystalline glucosamine sulfate to standard treatment, regardless of the stage of osteoarthritis, results in cost-effective outcomes within the acceptable willingness-to-pay threshold in Vietnam. Moreover, this study highlights that the early supplementation of glucosamine leads to a significant reduction in the Incremental Cost-Effectiveness Ratio (ICER) and substantially improves the cost-effectiveness of joint degeneration treatment compared to supplementation at later stages.

## Figures and Tables

**Figure 1 healthcare-11-02502-f001:**
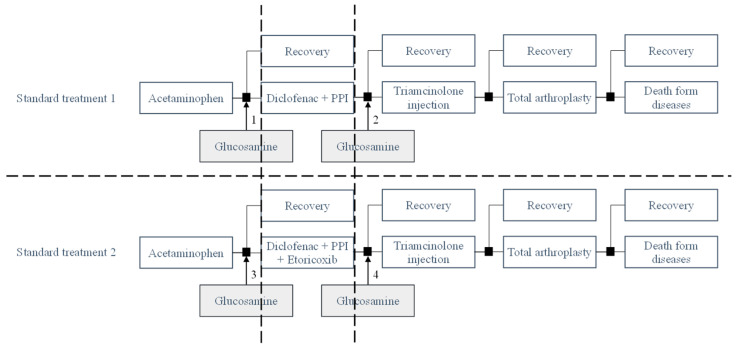
Model structure.

**Figure 2 healthcare-11-02502-f002:**
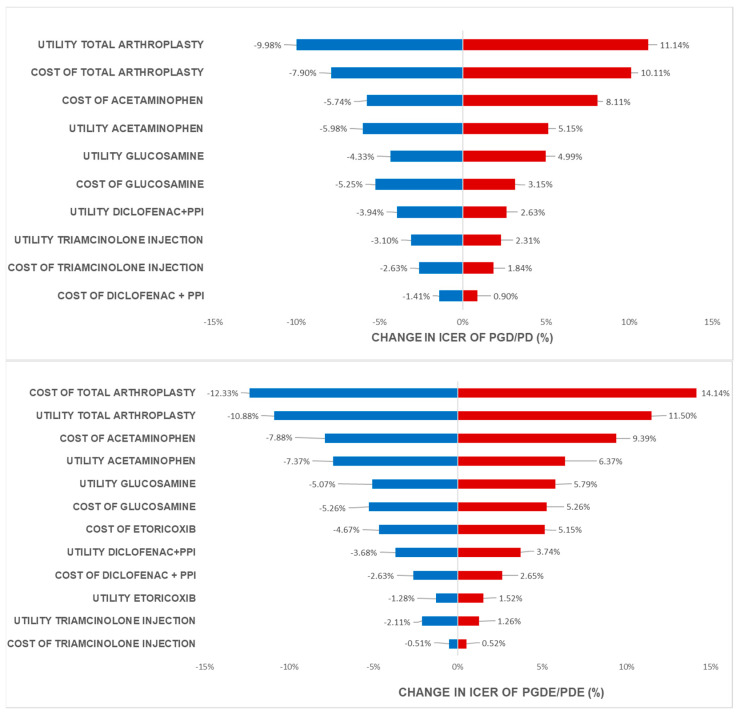
Tornado diagram of ICER change in different regimens when glucosamine was added before NSAIDs use.

**Figure 3 healthcare-11-02502-f003:**
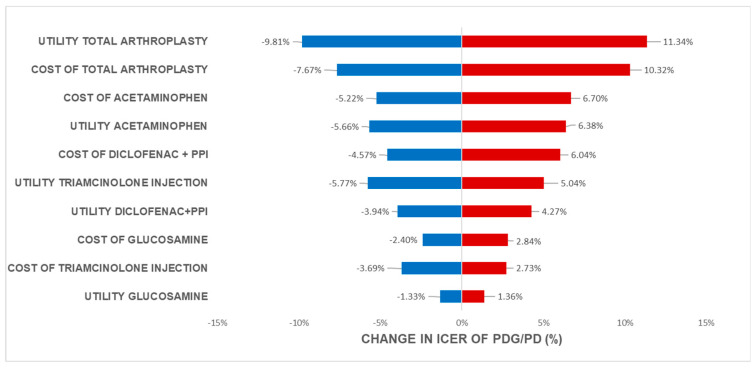
Tornado diagram of ICER change in different regimens when glucosamine was added after NSAIDs use.

**Table 1 healthcare-11-02502-t001:** Input data.

Component	Estimate	Sources
1. Transition probabilities		
Acetaminophen	0.3380	Herrero-Beaumont et al. (2007) [35]
Diclofenac + PPI	0.8647	Zacher et al. (2003) [36]
Triamcinolone injection	0.0392	McAlindon et al. (2017) [37]
Total arthroplasty	0.5122	Fortin et al. (1999) [38]
Etoricoxib	0.5657	Cannon et al. (2006) [39]
Crystalline glucosamine sulfate	0.3857	Herrero-Beaumont et al. (2007) [35]
2. Costs—VND (USD)		
Costs of treatment		
Acetaminophen (3000 mg/day) 500 mg—6 times/day	588,600	Vietnam Drug Administration, Ministry of Health, Vietnam [40]
	(25.06)	
Diclofenac (150 mg/day) 50 mg—3 times/day	60,210	Vietnam Drug Administration, Ministry of Health, Vietnam [40]
	(2.56)	
Omeprazole (20 mg/day) 20 mg—1 times/day	344,700	Vietnam Drug Administration, Ministry of Health, Vietnam [40]
	(14.68)	
Etoricoxib (60 mg/day) 30–60 mg—1–2 times/day	785,700	Vietnam Drug Administration, Ministry of Health, Vietnam [40]
	(33.45)	
Crystalline glucosamine sulfate (1500 mg/day) 500 mg—3 times/day	186,300	Vietnam Drug Administration, Ministry of Health, Vietnam [40]
	(7.93)	
Triamcinolone injection (40 mg every 3 months)—80 mg/2 mL	19,400	Vietnam Drug Administration, Ministry of Health, Vietnam [40]
	(0.83)	
Total Knee Arthroplasty (TKA)	88,712,500	Decree 39/2018/TT-BYT, Ministry of Health, Vietnam [41]
	(3777.25)	
Direct non-medical costs (VND)		
Food costs per person (1 month)	2,700,000	Research Center for Employment Relations (ERC) [42]
	(114.96)	
Transportation costs per person (1 month)	2,953,620	Wilbert B. et al. (2005) [43]
	(125.76)	
Indirect costs		
Cost of absenteeism (1 month-absent for Knee Joint Recovery Post-Surgery)	7,342,373	Hien Thu Trinh et al. (2022) [44]
	(312.63)	
3. Utilities		
Utility Acetaminophen	0.7010	Latimer et al. (2009) [45]
Utility Diclofenac + PPI	0.7230	Latimer et al. (2009) [45]
Utility Etoricoxib	0.7230	Latimer et al. (2009) [45]
Utility Glucosamine	0.6760	Olivier Bruyère et al. (2019) [46]
Utility Triamcinolone injection	0.6400	Losina et al. (2013) [47]
Utility total knee arthroplasty	0.7600	David Feeny et al. (2004) [48]

**Table 2 healthcare-11-02502-t002:** Total costs, health outcomes and Incremental Cost-Effectiveness Ratio.

	COST VND (USD)	QALYs
Standard treatment 1 (PD)	314,758,471 (13,401.96)	56.9374
Standard treatment 1 + glucosamine before NSAIDs (PGD)	348,133,295 (14,823.01)	100.7445
Standard treatment 1 + glucosamine after NSAIDs (PDG)	339,027,398 (14,435.30)	92.7119
ICER_PGD/PD_ VND (USD) per QALYs	761,858 (32.44)	
ICER_PDG/PD_ VND (USD) per QALYs	678,386 (28.88)	
ICER_PGD/PDG_ VND (USD) per QALYs	1,133,615 (48.27)	
Standard treatment 2 (PDE)	367,478,671 (15,646.71)	129.8038
Standard treatment 2 + glucosamine before NSAIDs (PGDE)	443,969,715 (18,903.59)	230.6720
Standard treatment 2 + glucosamine after NSAIDs (PDEG)	425,013,733 (18,096.47)	211.3288
ICER_PDE/PD_ VND (USD) per QALYs	723,519 (30.81)	
ICER_PGDE/PDE_ VND (USD) per QALYs	758,326 (32.29)	
ICER_PDEG/PDE_ VND (USD) per QALYs	705,735 (30.05)	
ICER_PGDE/PDEG_ VND (USD) per QALYs	979,981 (41.73)	

## Data Availability

Data were collected through a literature review of previous studies.

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
