# Peer review of "Economic Evaluation of Glucosamine in Knee Osteoarthritis Treatments in Vietnam"

_healthcare, 2023, doi:10.3390/healthcare11182502_

Round 1

Reviewer 1 Report

59-60 with a higher incidence 59 rate among females compared to males in Ho Chi Minh city, Vietnam :

Do you have the incidence rate males : females?

113 without any severe accompanying conditions. : which one?

190 for this study is 40 years old and older. : could you give a range of age and/or mean age?

200 economic conditions as Vietnam or with countries similar to Vietnam : name these countries 

239 what about opioids ? do you use them in Vietnam ?

296 Incremental Cost-Effectiveness Ratio (ICER) : what about the difference of ICER between females and males? If no data I would added as a limitation

383 the early use of Glucosamin : when is it early or late? What do you mean? 

421 results in cost-effective outcomes within the acceptable willingness-to-pay threshold in 421 Vietnam : your data also comes even from other countries how are you sure that the therapy would be effective and compatible with the Vietnamese population?

Author Response

Dear Reviewer 1,

Thank you very much for your comments and suggestions. Please see the attachment.

Best regards,

Reviewer 2 Report

This study explores the "Economic Evaluation of Glucosamine in Knee Osteoarthritis Treatments" through relevant data analysis. It is an intriguing research endeavor; however, there are some areas in the texts that require further improvement.

Introduction

the authors highlighted that "the clinical effectiveness of glucosamine, as demon strated in published research studies, remains inconclusive, leading to significant debates and difficulties in decision-making by reimbursement agencies", However, the author did not specify which studies support the clinical effectiveness of glucosamine, which studies do not support it, and it is recommended that the author could provide a list of relevant studies and references.

Materials and participants

The author should provide a detailed explanation of the calculation method and formula for ICER (Incremental Cost-Effectiveness Ratio).

Discussion

The title of this paper emphasizes the "Economic Evaluation of Glucosamine in Knee Osteoarthritis Treatments." However, in the discussion section, the author spent a considerable portion emphasizing that the treatment sequence could affect the treatment outcomes of patients. While this discussion is meaningful, the focus should be more on the "Economic Evaluation of Glucosamine." Although there is a paragraph comparing the results of this study with previous research, it falls short. More comprehensive discussion is needed.

The discussion section should also include an additional portion to explore the implications of the conclusions drawn from this study on the healthcare policy level.

Author Response

Dear Reviewer 2,

Thank you very much for your comments and suggestions. Please see the attachment.

Best regards,

Round 2

Reviewer 2 Report

The authors added references 27-29, I suggest the authors summarize the main ideas of references 27-29 and express them in the text.

Author Response

Dear Reviewer 02,

Thank you very much for your support. The author added information in line 95-97 with highlights. Please see the attached file,

Best regards,